# Ion-Mediated Self-Assembly of Graphene Oxide and Functionalized Perylene Diimides into Hybrid Materials with Photocatalytic Properties

**Maksim Sokolov, Alsu Nugmanova, Andrey Shkolin** (ID)**, Alexandra Zvyagina, Ivan Senchikhin** (ID) **and Maria Kalinina \*** (ID)

A. N. Frumkin Institute of Physical Chemistry and Electrochemistry, Russian Academy of Sciences, Moscow 119071, Russia
\* Correspondence: kalinina@phyche.ac.ru; Tel.: +7-(495)-955-46-80

**Abstract:** A novel ion-mediated self-assembly method was applied for integration of graphene oxide (GO), propanoic- and glutaric-substituted perylenes (glu-PDI and PA-PDI), and Zn $(OAc)_2$ into new hybrid materials with photocatalytic properties. The structuring of chromophores through coordination bonding on the GO surface is controlled by the chemistry of the PDI linkers. Four-substituted glu-PDI forms consolidated microporous particles, whereas di-substituted PA-PDI binds with GO into a macroporous gel-like structure. The GO/PDI controls without $Zn^{2+}$ ions form only non-integrated dispersions. Both hybrids can initiate photodestruction of 1,5-dihydroxynaphtalene (DHN) due to the effective charge separation between the PDI components and GO by generating hydroxyl radicals determined by luminescent probing with terephthalic acid. The reduction mechanism of photodegradation was confirmed by MALDI-TOF spectroscopy. The structure of the hybrids controls the rate of photodegradation process. The glu-PDI-based photocatalyst shows a smaller rate of photoreduction of $3.3 \times 10^{-2}$ min$^{-1}$ than that with PA-PDA ($4 \times 10^{-2}$ min$^{-1}$) due to diffusion limitations. Our results suggest that the ion-mediated synthesis is a useful and rational alternative for the conventional synthesis of GO-based functional hybrid materials through aromatic stacking between the graphene oxide and organic chromophores to produce new affordable and efficient photocatalysts.

**Keywords:** heterogeneous photocatalysts; photocatalysis; hybrid materials; graphene oxide; perylenes

## 1. Introduction

The design of new materials with photocatalytic properties addresses one of the most significant "grand challenges" in the 21st century, caused by continuously increasing technogenic emissions due to growing energy and resource consumption in the course of approaching the global demographic peak [1]. Increasing pollution as well as a decrease of the availability of non-renewable raw materials are demanding for the transition to the economy exploiting new, easily accessible, and sustainable energy resources. The use of solar energy represents an environment-friendly and low-cost pathway for reducing emissions from burning fossil fuels for chemical processes in industry and households [2,3].

In this context, the development of new easily available, low-cost materials for heterogeneous photocatalysis in aqueous media is of particular importance. These kinds of catalysts make it possible to regenerate natural water sources from synthetic pollutants in a fashion, which is non-disturbing for aquatic biota [4]. Heterogeneous photocatalysts offer an alternative to traditional antiseptic and antibacterial agents for industrial and domestic treatment [5] and can also be useful for chemical synthesis in water [6,7], the most eco-friendly solvent known.

Hybrid photocatalysts represent a new class of materials for heterogeneous photocatalysis [8]. These materials comprise photocatalytic activity of organic chromophores and

inorganic semiconductors, enhancing it or displaying it by new mechanisms that are not typical for the individual components [9,10].

Enhancement of photocatalytic activity is achieved through photoinduced charge separation with an appropriate selection of energy levels of the chromophore-semiconductor pair [11,12].

The integration of inorganic and organic components in the same material also allows broadening of its visible absorption spectrum as well as the enhancement of its photochemical stability [13]. The majority of hybrid photocatalysts based on these principles are those using titanium oxide, a cheap and accessible mineral [14]. The TiO2 hybrids are most commonly obtained through impregnating powder dispersions [10] or modifying the surface of TiO2 with the subsequent grafting of chromophore compounds, mainly polyaromatic [15] and macroheterocyclic [16,17] ones, through the formation of self-assembled monolayers on the particle surface.

Recently, there has been increasing interest in two-dimensional semiconductor materials as components of hybrid photocatalysts, such as graphene derivatives [18] (graphene oxide [19] and reduced graphene oxide [20]), bismuth tungstate [21,22], molybdenum sulfide [23], and MXenes [24,25] as the components.

Two-dimensional materials have several advantages over titanium oxide. The most important ones are the optical transparency in the visible range [26] and a relatively low Fermi level [27], which promotes the most efficient charge separation. Graphene derivatives, such as graphene oxide and reduced graphene oxide, offer the best combination of these characteristics [28,29].

Hybrid materials based on graphene oxide and poly- and heterocyclic chromophores such as porphyrins, phthalocyanines, and perylenes, are commonly produced by impregnating dry powders of carbon materials [30] and co-precipitation from solutions [31,32]. Component integration is achieved through aromatic stacking between the electron system of the chromophore and the $sp^2$-hybridized carbon carcass [30–32]. These interactions are relatively weak to provide stability as well as to control phase separation in the course of the synthesis [33]. Moreover, aromatic stacking often leads to complete quenching of a triplet state of the chromophore, thereby significantly reducing its potential photocatalytic activity [30,31].

The approach, which exploits covalent modification of GO by appropriately functionalized chromophores, provides mechanical stability and effective charge separation in a controlled manner [34]. However, this method is experimentally elaborated using a multistep procedure with the GO pre-functionalisation [35], which significantly increases the cost of resulting materials. The number of covalently grafted chromophores per thousand carbon atoms is comparatively low, limiting the usability of such materials.

We have recently shown that ion-driven non-covalent assembly can be used successfully for fabricating stable hybrid photocatalysts through coordination bonding between zinc porphyrinates and GO in Pickering emulsions [36,37]. Zinc acetate metal clusters were used as bonders, forming an anchoring layer on the GO surface to bind with carboxylic or pyridyl functional groups of porphyrin derivatives. These interactions led to the formation of structured surface-attached metal-organic framework (SURMOF) clusters between the GO sheets at the oil/water interface. The microporosity of such materials could be controlled by changing the length and type of substituents in the porphyrin ring. We showed that such materials are stable in aqueous media and can work as highly efficient heterogeneous ambivalent photocatalysts for the degradation of organic pollutants [37]. In the presence of oxygen, the photooxidation process proceeds mainly due to the release of a singlet oxygen, because porphyrins in SURMOFs retain the ability to transition to the triplet state upon a light irradiation of the hybrid material.

Although porphyrin complexes are certainly powerful photosensitizers showing high efficiency in hybrids, the synthesis of these compounds is, however, difficult. These linkers still remain expensive and not widely available, so their commercial potential in hybrid materials is questionable. Therefore, the ion-driven synthesis of hybrid photocatalysts needs

to be adapted to more affordable and commonly applied chromophores for expanding its potential applications.

In this work, we show for the first time that this strategy of ion-driven self-assembly can be extended toward the fabrication of new GO-based hybrid materials based on the derivatives of perylene bis-imide, a well-known chromophore widely used in organic electronics [38] and also in composite materials with photocatalytic properties [39]. Two perylene derivatives (*N,N′*-di(propanoic acid)-perylene3,4,9,10-tetracarboxylic diimide (PDI-PA) and *N,N′*-di(glutaric acid)-perylene-3,4,9,10-tetracarboxylic diimide (glu-PDI)) with carboxylic and glutamic substituents were used to assess the influence of the structure of the side substituents on the efficiency of integration. The obtained materials were characterized using a set of methods including optical and fluorescent microscopy, SEM, EDX, X-ray diffraction, and Raman spectroscopy. The photocatalytic activity in photodestruction of DHN of the materials was studied by UV-vis spectroscopy and by MALDI-TOF mass-spectrometry. The GO-PDI composites obtained by a conventional co-precipitation of GO and PDI chromophores from solution, in which the interaction of the components is possible mainly through aromatic stacking, were used as control systems to show the advantages of the ion-bonded hybrid materials and their better functioning as photocatalysts.

## 2. Materials and Methods

### 2.1. Materials

Triethylamine (TEA), 3,4,9,10-perylenetetracarboxylic dianhydride, glutamic acid, propionic acid, dihydroxynaphthalene (DHN), and zinc acetate (Zn (OAc)$_2$), all of analytical grade, were obtained commercially from Sigma-Aldrich and used without further purification. *n*-Hexane was obtained from ChemReactive (Russia) and purified by distillation over CaH$_2$. Water deionized to 0.2 mS·cm$^{-1}$ conductivity (pH 5.5 ± 0.1) was prepared using the "Vodoley" system (Russia). GO hydrosol (pH 4.2 ± 0.1) was synthesized from exfoliated graphite powder by a modified Hummers method described in [40] (the thickness of GO sheets is about 1.5 nm). *N,N′*-di(glutaric acid)-perylene-3,4,9,10-tetracarboxylic diimide (glu-PDI) and *N,N′*-di(propanoic acid)-perylene 3,4,9,10-tetracarboxylic diimide (PA-PDI) were synthesized by the methods described in [41] and [42], respectively.

The solutions of glu-PDI and PA-PDI with a concentration of $1.4 \times 10^{-5}$ M (pH 7.6) were prepared by dissolving 2.3 mg of glu-PDI or 1.8 mg of PA-PDI in 250 mL of deionized water containing 10 μL of TEA and then incubating in an ultrasound bath at 60 °C for 5 h. These solutions of glu-PDI and PA-PDI, GO hydrosol with a concentration of 2.5 g·L$^{-1}$ and Zn (OAc)$_2$ solutions in water with a concentration of 0.05 M, were used for preparation of the hybrid materials.

### 2.2. Synthesis of the GO/Zn (OAc)$_2$/PDI Hybrid Materials

The 2.5% *w/w* aqueous solution of graphene oxide (8 mL) with the GO size of 5 μm and the 0.5 mL aqueous solution of Zn (OAc)$_2$ ($5 \cdot 10^{-2}$ M) were mixed in a sealed vial and ultrasonicated for 20 min. Then, 33 mL of aqueous solution PA-PDI or glu-PDI were added into the vial; the mixture was shaken and placed into the oven for 24 h at 70 °C. The obtained powder was filtered, washed with water and methanol several times, and dried overnight under vacuum. Black powder was collected.

### 2.3. Synthesis of the GO/PDI Composites

The 2.5% *w/w* aqueous sol of graphene oxide (33 mL) with the GO size of 5 μm and $1.4 \cdot 10^{-5}$ M aqueous solution of PA-PDI or glu-PDI (8 mL) were mixed in a sealed vial, ultrasonicated for 20 min, and placed into the oven for 24 h at 70 °C. The obtained black powder was filtered, washed with water and methanol several times, and dried overnight under a vacuum.

### 2.4. Characterization of Perylene/GO Hybrid Materials

2.4.1. UV-Vis and Fluorescence Spectroscopy

UV-Vis spectra of solutions were recorded in 1 cm quartz cells with a two-beam spectrophotometer JASCO V-760 in the range 200–600 nm (path length is 10 mm). The fluorescence spectra were recorded using a spectrofluorometer JASCO FP-8350.

2.4.2. Optical and Fluorescence Microscopy

Microscopy images of the emulsion droplets and resulting materials were taken with a Lomo Mikmed-2 microscope at $20\times$ magnification equipped with an Olympus XC50 camera. Fluorescent images were taken under excitation with a mercury lamp DRS100, equipped with a 520–560 nm excitation bandpass filter and a 500–700 nm emission filter.

2.4.3. Raman Spectroscopy

Raman spectra were obtained with a Renishaw inVia Reflex Microscope system equipped with a Peltier-cooled CCD. The 532 nm line of a Nd:YaG laser was used for excitation. A laser light was focused on the sample through a $50\times$ objective to a spot size of 2 μm. The power on the sample was <0.02 mW. The spectral resolution was about 2 cm$^{-1}$.

2.4.4. X-ray Powder Diffraction (XRD)

X-ray diffraction (XRD) patterns were obtained using an Empyrean (Panalytical) diffractometer equipped with a 1-D position-sensitive X'Celerator detector. Ni-filtered Cu Kα-radiation was employed. Standard Bragg–Brentano (reflection) geometry was employed, allowing the acquisition of out-of-plane diffraction.

2.4.5. Scanning Electron Microscopy (SEM) and Energy-Dispersive X-ray Spectroscopy (EDX)

The sizes and morphology of perylene/GO hybrid materials were determined by scanning electron microscopy (SEM) using a Quanta 650 FEG microscope (FEI, Eindhoven, The Netherlands) equipped with an Octane Elect Plus energy dispersive X-ray (EDX) detector (EDAX, Pleasanton, CA, USA). To prepare the samples for analysis, the powders were deposited on an aluminum holder using double-sided conductive carbon tape, then placed in the instrument chamber and studied in a high vacuum at an accelerating voltage of 2 kV. X-ray spectral analysis was performed at an accelerating voltage of 30 kV. The spectra were analyzed by the original EDAX Genesis software (EDAX, Pleasanton, CA, USA).

2.4.6. BET Nitrogen Adsorption/Desorption Measurements

BET nitrogen adsorption/desorption measurements. The measurements of the specific surface area of the hybrid materials were carried out by the low-temperature nitrogen (77K) adsorption using an Autosorb IQ (Quantachrome Instruments, Boynton Beach, FL, USA). Before the analysis, the samples were degassed in a vacuum flow at 30 °C for 24 h. The specific surface area ($S_{BET}$) for the samples was calculated using the Brunauer–Emmett–Teller model (BET) at 5 points in the partial pressure range of 0.05–0.2.

2.4.7. Thermogravimetric Analysis (TGA)

Thermal stability of MoS2 nanosheets was evaluated by thermogravimetric analysis (TGA) with an SDT Q600 instrument (TA Instruments, New Castle, DE, USA). The experiments were carried out in open crucibles under nitrogen flow in a temperature range of 25–700 °C at a heating rate of 10 °C/min.

### 2.4.8. Matrix-Assisted Laser Desorption/Ionization Time-of-Flight (MALDI-TOF) Mass-Spectrometry

Matrix-assisted laser desorption/ionization time-of-flight mass-spectra were acquired with an Ultraflex Bruker Daltonics mass-spectrometer in the positive ion mode with 20 mV at the target. Aqueous solutions of dihydroxynaphthalene (DHN) were analyzed without matrices.

### 2.5. Photodegradation of 1,5-Dihydroxynaphthalene

For the investigation of photocatalytic activity of the perylene/GO powder, an aqueous solution (2.5% MeOH) of 1,5-dihydroxynaphthalene with a concentration of $1 \times 10^{-4}$ M was used. An amount of 5 mg of perylene/GO powder was added in 4 mL of the solution. The vial was sealed, shaken, and stored in darkness overnight. Then, the vial was irradiated with a mercury lamp and equipped with a yellow cut-off filter (>420 nm) without stirring. To analyze photodegradation rate, the mixture was centrifuged for 1 min, and 0.1 mL aliquot was taken, diluted up to 2 mL, and analyzed with a UV-Vis spectrometer.

## 3. Results and Discussion

### 3.1. Synthesis and Characterization

To obtain the perylene/GO hybrids in a powdered form, we applied a one-pot synthesis of these materials under hydrothermal conditions. The oxidized groups on the GO sheets promote the adsorption of metal clusters followed by an anchoring of perylenes to the surface of 2-D carbon (Figure 1) [43–45].

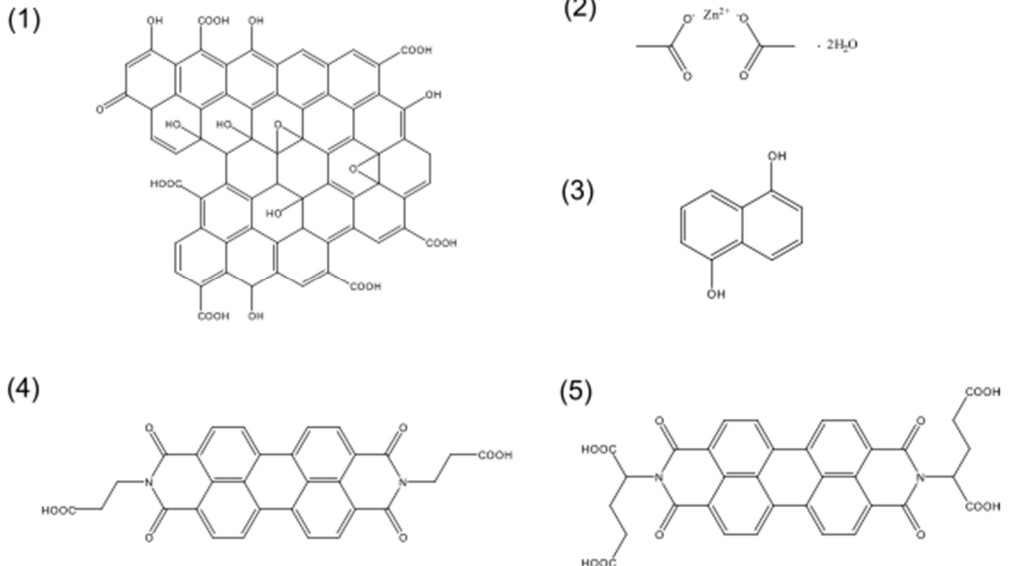

**Figure 1.** Structures of components of the GO/Zn (OAc)$_2$/PDI hybrids: (**1**) graphene oxide, (**2**) zinc acetate dihydrate, (**3**) 1,5-dihydroxynaphthalene, (**4**) PA-PDI, (**5**) glu-PDI.

The GO sols with an average size of the GO sheets of about 5 μm were first mixed with Zn (OAc)$_2$ and then with the PDI derivatives in water, kept at 70 °C for 24 h, then washed subsequently with water and ethanol to remove the excess of non-integrated salt and chromophores, and then dried under vacuum to remove water (Figure 2).

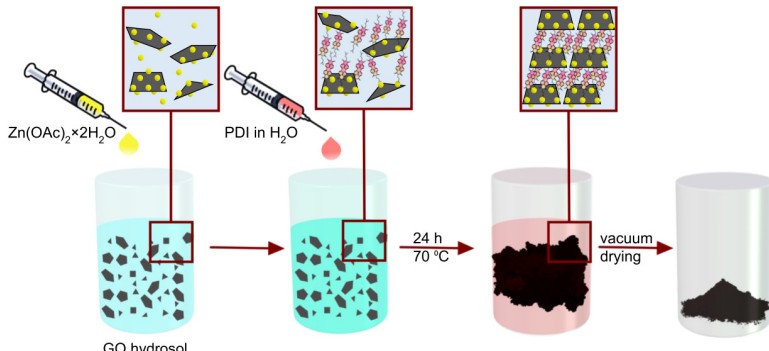

**Figure 2.** Schematically illustrated fabrication procedure for the formation of GO/Zn (OAc)$_2$/PDI hybrid materials.

The black powders (Figure 3A) obtained from the emulsions by a freeze-drying method exhibited strong characteristic emissions, indicating the integration of the perylenes with GO (Figure 3A, inset).

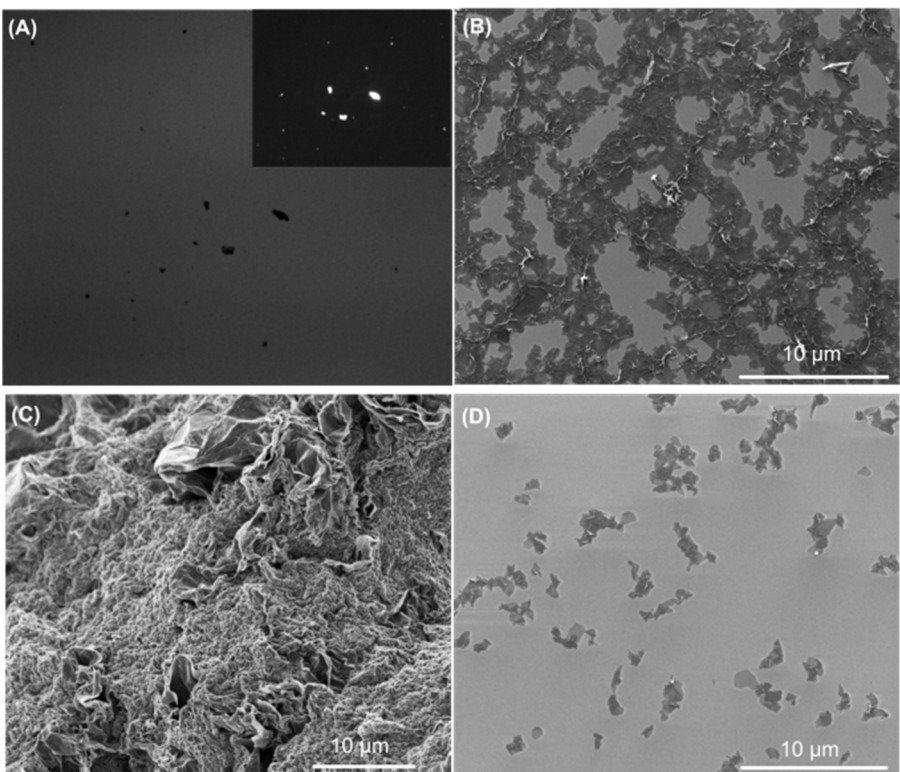

**Figure 3.** (**A**) Optical and (a, inset) fluorescent microscopy images of dried GO/Zn (OAc)$_2$/PA-PDI hybrid material. SEM microphotographs of (**B**) GO/Zn (OAc)$_2$/PA-PDI hybrid material, (**C**) GO/Zn (OAc)$_2$/glu-PDI hybrid material, and (**D**) GO/PA-PDI composite.

The morphology of the samples prepared by a drop-casting of the non-dried materials onto the solid supports followed by drying in the air was studied by the SEM method. The GO/Zn (OAc)$_2$/PA-PDI sample shows a gel-like morphology (Figure 3B), whereas the GO/Zn (OAc)$_2$/glu-PDI hybrid possesses a consolidated sponge-like structure (Figure 3C). We did not observe any indications of phase separation such as the formation of bulky crystallites of the PDI chromophores or striking GO fraction in both hybrids. The EDX data (Figure S1) confirm the presence of zinc in both GO/Zn (OAc)$_2$/PA-PDI and GO/Zn (Oac)$_2$/glu-PDI hybrids, suggesting the binding of zinc acetate metal clusters to the components of hybrid materials.

Figure 3D shows a typical morphological picture of the control GO/PA-PDI sample obtained without zinc acetate. Both controls are the dispersions of non-integrated sheets of GO. However, the GO/glu-PDI control also contains some bulky crystallites, indicating the phase separation of components during the preparation of the material through a conventional protocol exploiting only an aromatic stacking between GO and the chromophore (Figure S2). The results suggest that the molecular structure of the PDI derivatives determines the morphology of the resulting hybrid materials, because a number of functional carboxylated groups controls the ability of the molecule to bind to the GO matrix. For the ion-mediated binding through the formation of coordination bonds with zinc metal clusters, four anchoring groups of glu-PDI assist the aggregation of the GO sheets in a 3D space to form a percolated structure of GO/Zn (OAc)$_2$/glu-PDI. The PA-PDI presenting only two symmetrically attached carboxylic moieties cannot provide enough binding sites to induce percolation of the formed network-like structure of GO/Zn (OAc)$_2$/PA-PDI, which remains stable until drying. For the controls formed through aromatic staking of perylene cores with the non-oxidized portions of the GO matrix, the structure of glu-PDI is unfavorable for strong interactions due to the steric effect of the glutamic chains. Most likely, this chromophore interacts with GO predominantly through a hydrogen bonding between carboxylic groups of GO and those of glu-PDI. The strength of these interactions is too small to provide stability of the material in the aqueous media.

The ratios of organic/inorganic components in the GO/Zn (OAc)$_2$/PA-PDI and GO/Zn (OAc)$_2$/glu-PDI hybrids were determined by TGA analysis [46] using the procedure reported earlier for the ZnTCPP-based SURMOF in the layered double hydroxide matrix and porphyrin-based GO hybrids [45]. The average relative mass amount of the PDI components calculated from the TGA was 19% for GO/Zn (OAc)$_2$/PA-PDI and 18% for GO/Zn (OAc)$_2$/glu-PDI (Figure S3). We note especially that the ion-mediated hydrothermal synthesis allows accumulation of comparatively large amounts of perylene derivatives in the resulting hybrids, suggesting that this kind of binding exploits strong interactions between the components [37]. The zinc-free controls showed unstable thermal behavior. For GO/PA-PDI and GO/glu-PDI, the calculated average relative mass amount of the PDI components was 19 and 14%, respectively (Figure S3). This difference correlates with the smaller efficiency of the integration of four substituted glu-PDI with GO than that for PA-PDI.

Figure 4 shows the XRD patterns of the GO/Zn (OAc)$_2$/PA-PDI and GO/Zn (OAc)$_2$/glu-PDI hybrid materials (curves 1 and 2, respectively) and those of GO/PA-PDI and GO/glu-PDI controls (curves 3 and 4, respectively). The structural picture for all synthesized materials suggests that the PDI derivatives do not form bulk crystal phases, as is evidenced by the absence of peaks at small angles. All resolved peaks are observed in the range of 20–30 2θ degrees corresponding to the diffraction of carbon carcasses aggregated by aromatic stacking. The particular position of peaks depends on the degree of aggregation, which is determined by the structure of aromatic molecules. The distinct peaks at 21 and 24 degrees appear in the XRD patterns of GO/Zn (OAc)$_2$/PA-PDI and GO/Zn (OAc)$_2$/glu-PDI hybrids as well as in those of controls. However, the XRD patterns of the GO/PA-PDI and GO/glu-PDI materials also present several peaks at higher angles, indicating a high degree of aromatic conjugation. The observed difference in the XRD pictures for the zinc-bonded hybrids and the controls suggests that zinc ions prevent strong interactions of the GO-adsorbed chromophore molecules through π-π stacking between the aromatic systems.

The Raman spectra of hybrid materials (Figure 5) show two broad bands of graphene oxide in the range of 1250–1700 cm$^{-1}$. In the same range, characteristic bands of the perylene core are observed: 1303, 1377, 1457, and 1573 cm$^{-1}$. The bands in the range of 2500–3000 cm$^{-1}$ correspond to the second-order vibrations of both graphene oxide and perylenes. The presence of pronounced overtones and the significantly higher intensity of the main vibrational bands of the perylene molecules, compared with the Raman spectra of the original perylenes (Figure S4), indicates the enhancement of the Raman signal due to the interaction of graphene oxide and perylene.

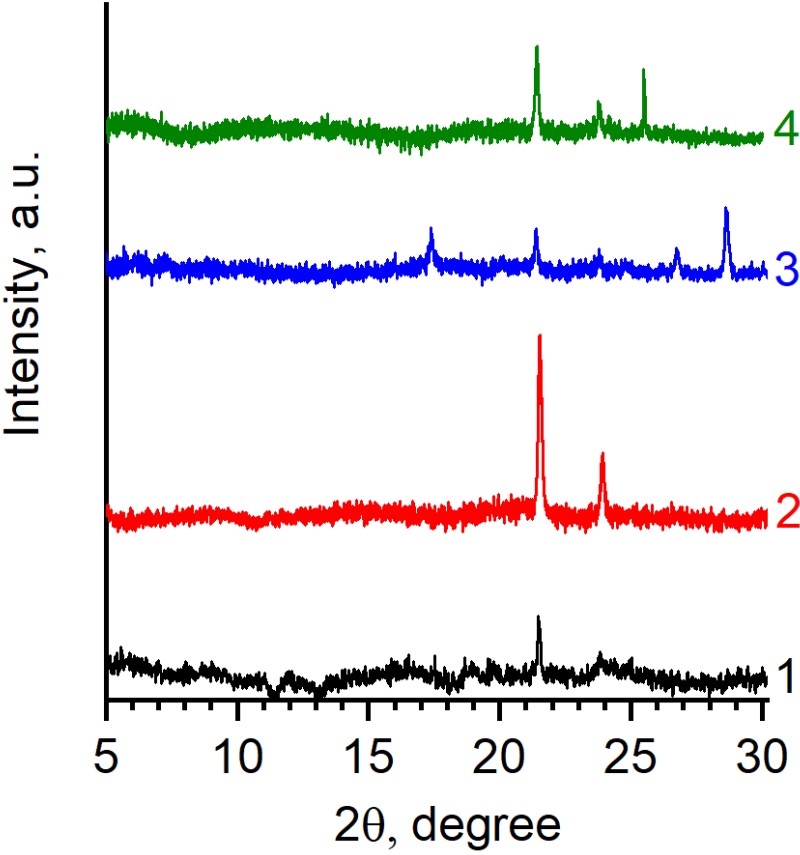

**Figure 4.** Experimental XRD patterns of (**1**) GO/Zn (OAc)$_2$/PA-PDI hybrid material, (**2**) GO/Zn (OAc)$_2$/glu-PDI hybrid material, (**3**) GO/PA-PDI composite, and (**4**) GO/glu-PDI composite.

The BET nitrogen adsorption method was applied for characterisation of the sorption properties of the obtained materials [47]. The BET adsorption-desorption isotherms of standard nitrogen vapor at 77 K for GO/PA-PDI control, GO/Zn (OAc)$_2$/glu-PDA, and GO/Zn (OAc)$_2$/PA-PDA hybrids are shown in Figures S5–S7, respectively. The isotherms of GO/PA-PDI control and GO/Zn (OAc)$_2$/PA-PDA in the coordinates a = $f$ (P/PS) have a form of type VI. The isotherm type is representative of layer-by-layer adsorption on a highly uniform surface that is typical for control graphene oxide alone. The isotherm GO/Zn (OAc)$_2$/glu-PDA has a form of type III with a hysteresis H1 [48]. Table 1 presents the parameters of the porous structure of the hybrid materials. The mesopore and macropore volume ($W_S$–$W_0$) of the GO/Zn (OAc)$_2$/PA-PDA is about 0.25 cm$^3$/g or 100% of the total pore volume, whereas micropores were not detected. The GO/Zn (OAc)$_2$/glu-PDA material contains a certain amount of micropores up to 0.06 cm$^3$/g, whereas the total pore volume reaches 0.26 cm$^3$/g. This hybrid thus contains a micro-mesoporous structure predominantly formed by meso- and macropores. This difference in the adsorption properties of the Zn$^{2+}$-bonded hybrids is in agreement with their morphologies observed by SEM and determined by the molecular structure of the PDI chromophores. The increase in the number of anchoring groups in glu-PDA with respect to those in PA-PDA leads to the emergence of microporosity, whereas the open gel-like morphology of the PA-PDI-based hybrid contains only meso-/macropores.

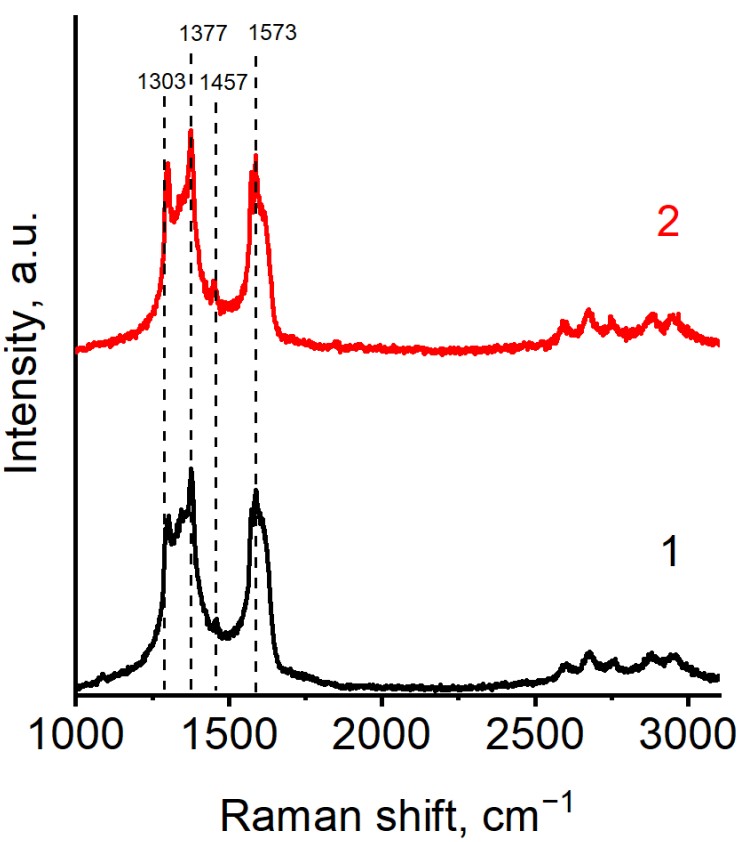

**Figure 5.** Raman spectra of (**1**) GO/Zn (OAc)$_2$/PA-PDI and (**2**) GO/Zn (OAc)$_2$/glu-PDI hybrid materials.

**Table 1.** Characterization of the GO/PDI-based materials by BET nitrogen adsorption method.

| Method → | BET | D-R | D-R | D-R | Kalvin | Kalvin |
|---|---|---|---|---|---|---|
| Sample | $^a$ $S_{BET}$, m$^2$/g | $^b$ $W_0$, cm$^3$/g | $^b$ $x_0$, nm | $^b$ $E_{0(N2)}$, kJ/mol | $^c$ $W_S$, cm$^3$/g | $^c$ $W_S$–$W_0$, cm$^3$/g |
| GO/PA-PDA | 40 | 0.02 | 1.03 | 3.85 | 0.06 | 0.04 |
| GO/Zn (OAc)$_2$/glu-PDI | 125 | 0.06 | 0.84 | 4.75 | 0.26 | 0.20 |
| GO/Zn (OAc)$_2$/PA-PDI | 190 | 0.00 | - | - | 0.25 | 0.25 |

$^a$—$S_{BET}$—specific surface area of the adsorbent determined by BET method (m$^2$·g$^{-1}$). $^b$—$W_0$—specific volume of micropores (cm$^3$·g$^{-1}$); $x_0$—half—width of the micropore (nm); $E_{0(N2)}$—characteristic adsorption energy of standard steam in terms of nitrogen (kJ·mol$^{-1}$); this was calculated by the Dubinin–Radushkevich (D-R) equation. $^c$—The Kelvin equation was used to calculate $W_S$—specific total pore volume (cm$^3$·g$^{-1}$); ($W_S$–$W_0$)—specific volume of meso and macropore (cm$^3$·g$^{-1}$).

### 3.2. Hybrid-Assisted Photodegradation Studies

To investigate the photocatalytic properties of the hybrid materials and the control samples, all the materials studied herein were preliminary equilibrated in the solution of DHN. The resulting decrease of the absorbance of DHN by 14–15% for both controls did not exceed that measured for the pure GO powder, suggesting that the interactions between the perylene chromophores and non-functionalized matrix, if they occurred, did not lead to the formation of a porous structure with a high adsorption ability. However, the GO/Zn (OAc)$_2$/glu-PDA showed the ability to absorb about 34% of the substrate, which confirms

the formation of a certain number of micropores in such material. After the adsorption was completed, the reaction vials were irradiated by a halogen lamp at λ > 410 nm under permanent stirring. The kinetics of photobleaching were measured by monitoring the changes in the characteristic adsorption of the substrates.

The dynamic spectral patterns of photodestruction of DHN are similar for both hybrid materials inducing a decrease of the characteristic absorption at 222 nm and a progressive increase of the characteristic band of 5-hydroxy-1,4-naphthoquinone (Juglone), a well-known product of the DHN oxidation, at 423 nm [49,50] (Figure 6). Another remarkable aspect of this reaction is that the spectral pictures of the destruction of DHN show the formation of the reaction product absorbing at 257 nm, that is, in a range typical for the absorption of phthalates, whose formation has also been observed during photoreduction in the porphyrin-based SURMOF/GO hybrids [36]. Similar spectral patterns were also observed for the controls, although with less pronounced changes in the optical picture (Figure S8a,b). We note especially that phthalates were not found in the controls of DHN solutions irradiated for 24 h without photocatalysts, and all the samples prepared for mass spectrometry were obtained in the Teflon vials; that is, the phthalates are not the impurities added to the samples in the course of the experiments.

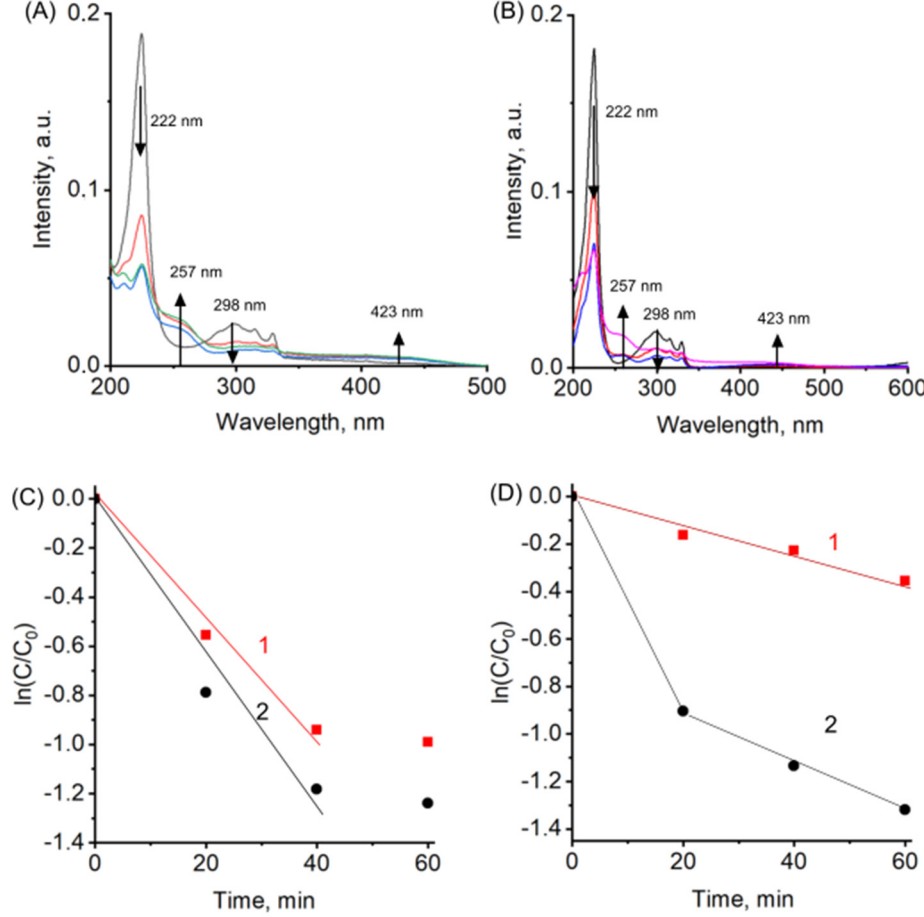

**Figure 6.** UV-Vis spectra of photodegradation of DHN in aqueous solution assisted by GO/Zn (OAc)$_2$/PA-PDI hybrid materials (**A**) and the GO/PA-PDI composite (**B**) at λ > 410 nm, and the corresponding kinetic curves of (**C**) GO/Zn (OAc)$_2$/glu-PDI (1) GO/Zn (OAc)$_2$/PA-PDI (2) hybrid materials and (**D**) GO/glu-PDI (1) GO/PA-PDI (2) composites. The kinetic curves were calculated from the peak at λ = 222 nm. Arrows indicate the reaction time. The concentration of DHN was $10^{-5}$ M. The spectral picture of the reaction is given only for the PA-PDI-based materials; for that of glu-PDI, see Figure S8.

The reaction rate constants were calculated both by using the increments of the decrease of the DHN band at 222 nm and those of the increasing band of Juglone at 423 nm (Figure S8c,d). The relative conversion of the dye within a 60 min time interval in the presence of the GO/Zn (OAc)$_2$/PA-PDA and GO/Zn (OAc)$_2$/glu-PDA hybrid reached 63 and 57% min$^{-1}$, respectively. The rate constants calculated by the decrease of the concentration of the substrate were $1.9 \times 10^{-3}$ and $6 \times 10^{-4}$ min$^{-1}$ for GO/PA-PDI and GO/glu-PDI controls, respectively. For GO/Zn (OAc)$_2$/glu-PDI GO/Zn (OAc)$_2$/PA-PDI, the rate constants were $2.9 \times 10^{-2}$ min$^{-1}$ and $3.3 \times 10^{-2}$ min$^{-1}$, respectively. The values for hybrid materials are comparable to those reported for other potent GO-based materials with photocatalytic properties [37,51]. These results suggest that the binding through aromatic stacking cannot provide as many efficient interactions between GO and the chromophores as the coordination bonding does in the ion-bonded hybrid materials. However, the certain reproducibly measured difference in the efficiency between GO/Zn (OAc)$_2$/glu-PDI and GO/Zn (OAc)$_2$/PA-PDI despite similar mass loading of chromophores in these materials is an indication that the morphology of the heterogeneous catalysts as well as the accessibility of their surface for the substrate is one of the key factors determining their functional behavior. The open gel-like structure of the PA-PDI-based photocatalysts is more favorable for the surface-mediated photocatalytic reaction than that of the consolidated sponge-like glu-PDI-based material, whose photocatalytic activity is determined by diffusion limitations. The diffusion of both the starting substrate and its photodegradation products is reasonably less rapid in the consolidated microporous structure than in the open gel-like one. However, this difference may manifest itself the most dramatically for the diffusion of the reaction products.

MALDI-TOF analysis of the photocatalytic transformations of DHN after the reactions in the supernatant solutions revealed the peaks of various phthalates as well as the peaks of polymerized products, showing the similarity of the reaction pathway irrespective of the structure of the hybrid photocatalyst (Figure 7). This chemical picture is also similar to that which we have observed earlier for this reaction in the presence of porphyrin-based hybrid materials [37]. The photoinduced transformations of DHN assisted by the hybrids therefore proceeded through oxidation of the starting compound DHN and through the further reduction of the reaction products.

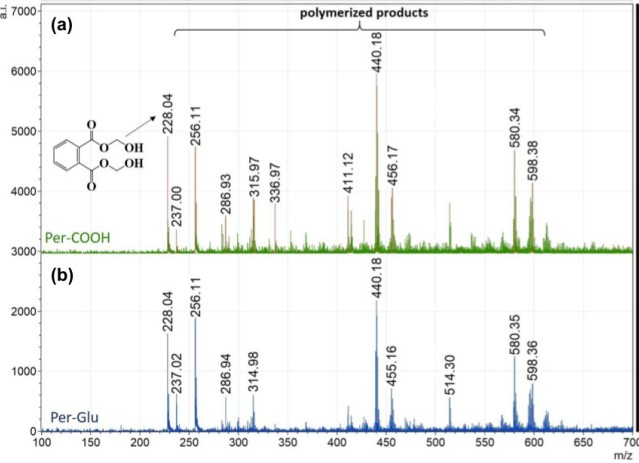

**Figure 7.** MALDI-TOF spectra of aqueous solutions of supernatants after the reaction of photodegradation of DHN assisted by the GO/Zn (OAc)$_2$/glu-PDI (**a**) and GO/Zn (OAc)$_2$/PA-PDI (**b**).

This mechanism exploits efficient charge separation between GO and the PDI derivatives, because the reductive destruction can occur only via direct transferring of photoexcited electrons from the perylene core to the GO substrate in combination with the generation of holes on the PDI centers to produce hydroxyl radicals. The formation of these intermediates in water in the presence of the PDI-based catalysts was confirmed

by the photoluminescence probing with terephthalic acid (TPA) as a sensing agent [52]. The solution of TPA was equilibrated with the hybrid materials for sorption within 12 h and then irradiated for 15 min. The intense luminescence at 410–450 nm is a specific indication of the formation of a monohydroxy terephthalate ion due to the interactions with hydroxyl radicals released by the photocatalysts (Figure 8). The emission of the probing TPA is remarkably lower over the GO/Zn (OAc)$_2$/glu-PDI than in the presence of GO/Zn (Oac)$_2$/PA-PDI, thereby confirming the dominating role of the photocatalyst structure in the photodestruction process.

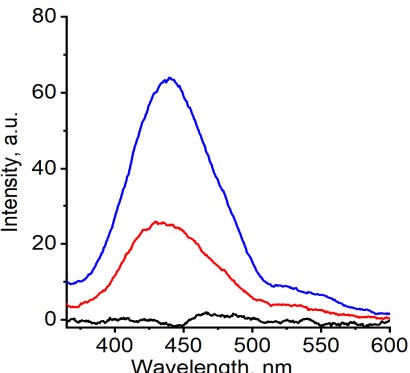

**Figure 8.** Fluorescence ($\lambda$ex = 312 nm) spectral patterns of the aqueous solution of TPA (black) forming TPA-OH after irradiation with visible light at 15 min in the presence of GO/Zn (OAc)$_2$/glu-PDI (blue) and GO/Zn (OAc)$_2$/PA-PDI (red).

Figure 9 shows the proposed mechanism for DNH transformations on the surface of the hybrid photocatalysts, suggesting the separation of the photoinduced charge on the PDI photoactive centers. The hole reacts with water, producing hydroxyl radicals, which, in turn, can react with DHN, yielding degraded products (Figure 9, 1 and 2). Further transformation including polymerization is a consequence of the synergy between GO with its low Fermi level and the structured PDI centers. The electron–hole recombination seems to be suppressed in the hybrid structure, providing good carrier mobility. The photogenerated electrons transferred on the GO sheet can react directly with the products of interactions with hydroxyl radicals (the process 3 in Figure 9) and initiate their dimerization and further polymerization.

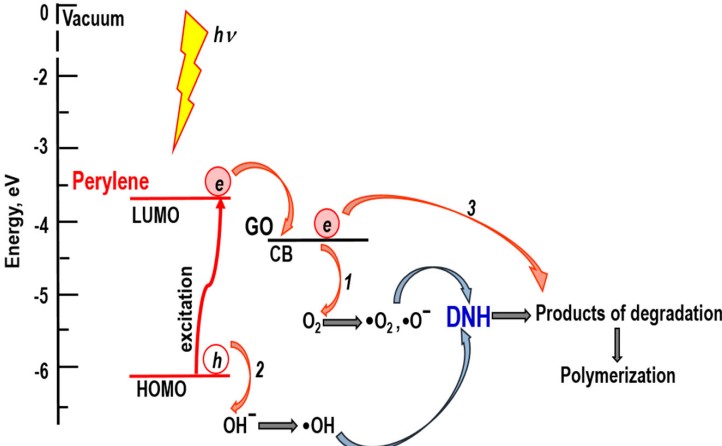

**Figure 9.** The proposed mechanism of the photoinduced destruction in the GO/Zn (OAc)2/PDI micropores. The conduction band edge (CB (−4.33 eV)) for GO and the HOMO energy level (−3.6 eV) for PDI are from Reference [53]. For DNH, the HOMO energy level (−5.5 eV) and optical bandgap (3.6 eV) for calculation of the LUMO level are from Reference [54].

## 4. Conclusions

In summary, we demonstrated that the ion-driven self-assembly of the functionalized chromophores such as perylene derivatives and graphene oxide is an effective alternative strategy for fabricating hybrid materials with photocatalytic properties. Unlike the conventional approach exploiting only aromatic stacking between the perylene core and the conjugated carbon carcass without appropriate control of the resulting structure, the integration of graphene oxide and chromophores through ion-mediated coordination bonding allows structuring of the components due to both the space-directed character of coordination bonds and the structure of the graphene oxide sheets presenting carboxylic groups predominantly at the boundaries of the carbon particles. The binding of carboxylic groups of perylene chromophores and those of the graphene oxide sheets through the interactions of metal ions or clusters such as zinc acetate leads to the formation of consolidated morphologies because of the integration of the GO particles through a network of coordination bonds. The particular kind of structuring depends on the molecular structure of the organic linker. We showed that the increase in the number of perylene substituents participating in coordination bonding promotes aggregation of the GO sheets into microporous aggregates because of the large number of binding sites, whereas the symmetrically functionalized chromophore presenting only two carboxylic moieties can form a gel-like structure that percolates throughout the available volume of solution of the components. Although both kinds of the as-formed hybrids showed photocatalytic activity with respect to model organic substrate, the gel-like structure shows better performance owing to the open and easily accessible morphology for the substrate molecules. The hybrid material with the microporous structure shows a somewhat smaller rate of photocatalytic transformation of the substrate, most likely due to diffusion limitations. Both kinds of hybrids exhibit the ability to initiate photodegradation of organic substrates through a charge separation between graphene oxide and perylene components in monomeric form. The mechanisms of the photodegradation involve both photooxidation in solution as well as photoreduction on the surface of the photocatalysts followed by polymerization of the products due to the donor-acceptor nature of graphene oxide.

The ion-mediated self-assembly is therefore a convenient supramolecular tool for modulating the structure and morphology of the GO-based hybrid materials comprising the chromophores, which are prone to aggregation, and for tuning their photocatalytic activity. We believe that our method might be extended toward the wide range of various chromophores for optimizing the combinations of hybrid components to achieve the best photocatalytic properties in an affordable fashion, saving energy and valuable chemicals.

**Supplementary Materials:** The following supporting information can be downloaded at: https://www.mdpi.com/article/10.3390/jcs7010014/s1.

**Author Contributions:** Conceptualization, M.K.; methodology, M.S., A.N., A.S., A.Z. and I.S.; validation, M.S., A.N., A.S., A.Z. and I.S.; formal analysis, A.N. and M.S.; investigation, M.S. and A.S.; resources, A.S., A.Z. and I.S.; data curation, M.S.; writing—original draft preparation, M.S. and M.K.; writing—review and editing, M.K.; visualization, M.S. and A.N.; supervision, M.K.; project administration, M.K.; funding acquisition, M.K. All authors have read and agreed to the published version of the manuscript.

**Funding:** This research was funded by the Russian Science Foundation, grant number 20-13-00279.

**Institutional Review Board Statement:** Not applicable.

**Informed Consent Statement:** Not applicable.

**Data Availability Statement:** The data presented in this study are available on request from the corresponding author.

**Acknowledgments:** Analytical measurements were performed using the equipment of CKP FMI IPCE RAS. The authors thank A. Averin for his assistance with Raman spectroscopy experiments.

**Conflicts of Interest:** The authors declare no conflict of interest.

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
