# Peer review of "Ion-Mediated Self-Assembly of Graphene Oxide and Functionalized Perylene Diimides into Hybrid Materials with Photocatalytic Properties"

_jcs, doi:10.3390/jcs7010014_

Round 1
Reviewer 1 Report
In this paper, the authors have used the ion-mediated self-assembly method to integrate the graphene oxide (GO), two types of perylenes (PDI-PA and PDI-glu) and zinc acetate (Zn(OAc)2) into new hybrid materials, and then studied the photocatalytic activity in the photodestruction of DHN of the obtained hybrid materials. Overall, this paper merits its publication in Journal of Composites Science. The reviewer recommends it to be accepted if the following comments could be properly addressed.
1. In this paper, Fig. 6c shows that the kinetic curve of GO/Zn(OAc)2/PA-PDI is approximately the same with that of GO/Zn(OAc)2/glu-PDI before 40min. But after 40min, it begins to drop more quickly, could the authors briefly explain this phenomenon?
2. In this paper, Figs. 6c and d show that the decrease of concentration of GO/PA-PDI composites seems closed to that of GO/Zn(OAc)2/glu-PDI hybrid materials in the 60 minutes. But the reaction rate constant calculated for GO/PA-PDI was much smaller than that for GO/Zn(OAc)2/glu-PDI, could the authors explain more how the reaction rate constants were calculated?
3. In this paper, the authors stated that the glu-PDI-based photocatalyst has a smaller rate of photoreduction than that of PA-PDI due to diffusion limitations. Could the authors explain this more detailly, especially which chemical species' diffusion would affect the most?
Author Response
Reviewer 1. We thank the Reviewer for a careful reading of our submission and valuable remarks.
Comment 1&2. In this paper, Fig. 6c shows that the kinetic curve of GO/Zn(OAc)2/PA-PDI is approximately the same with that of GO/Zn(OAc)2/glu-PDI before 40min. But after 40min, it begins to drop more quickly, could the authors briefly explain this phenomenon? In this paper, Figs. 6c and d show that the decrease of concentration of GO/PA-PDI composites seems closed to that of GO/Zn(OAc)2/glu-PDI hybrid materials in the 60 minutes. But the reaction rate constant calculated for GO/PA-PDI was much smaller than that for GO/Zn(OAc)2/glu-PDI, could the authors explain more how the reaction rate constants were calculated?
Response: We thank the Reviewer especially for noticing this inconsistence. This was an unintended mistake in typing upper symbols. In fact, the difference in the rate constant is indeed not that large. We reproduce the kinetic measurements twice with better time resolution to address this concern. There is no actually a sharp drop of the activity of the catalysts, both showed comparable activity, although that of the glu-PDI-based material was reproducibly smaller. The values of rate constants were recalculated for the accuracy of presentation. This fragment was corrected accordingly.
Comment 3. In this paper, the authors stated that the glu-PDI-based photocatalyst has a smaller rate of photoreduction than that of PA-PDI due to diffusion limitations. Could the authors explain this more detailly, especially which chemical species' diffusion would affect the most?
Response: We believe that the diffusion of both the starting substrate and its photodegradation products is reasonably less rapid in the consolidated microporous structure than in the open gel-like one. For polymerization products this difference may manifest itself because of certain accumulation of the product. This point was added to a corresponding paragraph in the discussion.
Reviewer 2 Report
Review report:
Authors reported “Ion-mediated self-assembly of graphene oxide and functionalized perylene diimides into hybrid materials with photocatalytic properties.”. The organization of this work is good, and the discussion is well organized. The characterization and calculation are both solid for the conclusion. Nevertheless, I have some comments which are listed below.
1. The synthesis scheme is not clear, and it should be revised in a detailed way.
2. The author claimed that their work is a novel investigation, however, myriads of works related to “glu-PDI-based photocatalyst “ have been published to date. So, authors should change the way of the presentation focusing on novelty. The introduction should be improved with a paragraph describing the novelty and importance of the work.
3. The authors must carefully claim their novelty in the INTRODUCTION. In addition, the authors need to do some formatting errors that should be carefully checked and corrected in the text.
4. The source and purity of all chemicals used should be specified. Authors should be looked at into below suggested references and can cite and take references regarding the “Source and Purity issues”: “Dalton Trans., 47 (2018), pp. 15545-15554”, “Nanomaterials, 2022, 12(22), 3982”, which references should be cited in your revised manuscript for better understanding.
5. A summary of key improvements compared to findings in the literature [provide a couple of references to indicate key improvements].
6. The surface area measurements are very important for active electrodes in supercapacitor applications. Please provide nitrogen adsorption and desorption (BET analysis) of the active materials (glu-PDI-based photocatalyst). Authors should be looked at into below suggested references with your work and cited them in your revised manuscript: “Nanomaterials, 2022, 12(18), 2330”, which given references should be cited in the revised manuscript.
7. The authors have missed their mass loadings of “GO/glu-PDI composite” electrode materials should be included in the revised manuscript. Authors should be looked at into below suggested references and can cite and take references regarding the mass loadings: “Nanomaterials, 2022, 12 (14), 3187”, which should be cited in your revised manuscript for better understanding.
8. Please provide the comparison table, which you need to prove that your material is superior to previously reported literature.
9. The reviewer also suggests that authors get professional English services to correct the grammatical error and refine the expressions in the body of the manuscript.
10. Authors should be trimmed/condensed the ‘Abstract’ and ‘Conclusion’ sections in the revised manuscript. Please keep highlights of the whole manuscript in both sections.
Author Response
Reviewer 2. We thank the Reviewer for a careful critical reading of our submission and helpful remarks assisting a lot the improvement of our submission.
Comment 1. The synthesis scheme is not clear, and it should be revised in a detailed way.
Response: The schematics were redrawn for clarifying the procedure and mechanism of the formation of hybrid materials in our work.
Comment 2&3. The author claimed that their work is a novel investigation, however, myriads of works related to “glu-PDI-based photocatalyst “ have been published to date. So, authors should change the way of the presentation focusing on novelty. The introduction should be improved with a paragraph describing the novelty and importance of the work. …The authors must carefully claim their novelty in the INTRODUCTION. In addition, the authors need to do some formatting errors that should be carefully checked and corrected in the text.
Response: Wil all due respect to the Reviewer’s expertise, we are afraid that we cannot agree with this assessment. We could not find any published reports on the photocatalysts with glutamate-substituted perylene diimide, that is, this kind of material is actually novel. We agree, that there are a number of PDI-based hybrids used for different photocatalytic reactions. Our goal was, however, to demonstrate the potential not of the particular PDI-based hybrid but rather the advantage of the method of ion-driven self-assembly we pioneered to apply for their fabrication. We revise the introduction section to emphasize this aspect of our work and its novelty. The changes we made are marked in green in the revised version.
Comment 4. The source and purity of all chemicals used should be specified. Authors should be looked at into below suggested references and can cite and take references regarding the “Source and Purity issues”: “Dalton Trans., 47 (2018), pp. 15545-15554”, “Nanomaterials, 2022, 12(22), 3982”, which references should be cited in your revised manuscript for better understanding.
Response: The necessary information regarding this issue was added to the Experimental section.
Comment 5&8. A summary of key improvements compared to findings in the literature [provide a couple of references to indicate key improvements]. Please provide the comparison table, which you need to prove that your material is superior to previously reported literature.
Response: To address the Reviewer’s concern, we added to Reference to ACS Appl. Mater. Interfaces 2016, 8, 44, 30225–30231https://doi.org/10.1021/acsami.6b10186 , in which the highest known rate constant for PDI-based materials as photocatalysts for destruction of phenolic aromatics was reported. Our values are comparable to this one confirming the high potency of our materials, which are largely more affordable than TCNQ-based ones. It is hardly possible to make up a table with full and adequate comparison within a short while, because all aspects such as concentrations, probe preparations, registration methods etc. needs to be taken into account.
Comment 6. The surface area measurements are very important for active electrodes in supercapacitor applications. Please provide nitrogen adsorption and desorption (BET analysis) of the active materials (glu-PDI-based photocatalyst). Authors should be looked at into below suggested references with your work and cited them in your revised manuscript: “Nanomaterials, 2022, 12(18), 2330”, which given references should be cited in the revised manuscript.
Response: To address the Reviewer’s remark, the BET nitrogen adsorption-desorption curves are given in the Supplementary Information as FigsS4-6. The calculated values of sorption parameters for hybrids materials as well as for control were summarized in Table 1 and added to the main text. The suggested references were added to the appropriate paragraph.
Comment 7. The authors have missed their mass loadings of “GO/glu-PDI composite” electrode materials should be included in the revised manuscript. Authors should be looked at into below suggested references and can cite and take references regarding the mass loadings: “Nanomaterials, 2022, 12 (14), 3187”, which should be cited in your revised manuscript for better understanding.
Response: The TGA curves for GO/PDI hybrids and controls were added to the Supplementary Information and discussed in the main text in the corresponding paragraphs. The suggested references were added to the appropriate paragraph.
Comment 9. The reviewer also suggests that authors get professional English services to correct the Response: The grammar and style were checked throughout the manuscript accordingly.
Round 2
Reviewer 1 Report
The authors have addressed my concerns in this revision.
Reviewer 2 Report
It can be accepted in its current format.